# The bile acid-sensitive ion channel is gated by Ca²⁺-dependent conformational changes in the transmembrane domain

Makayla M. Freitas[1] & Eric Gouaux [1,2] ✉

The bile acid-sensitive ion channel (BASIC) is the least understood member of the mammalian epithelial Na⁺ channel/degenerin (ENaC/DEG) superfamily of ion channels, which are involved in a variety of physiological processes. While some members of this superfamily, including BASIC, are inhibited by extracellular Ca²⁺ ($Ca^{2+}_o$), the molecular mechanism underlying Ca²⁺ modulation remains unclear. Here, by determining the structure of human BASIC (hBASIC) in the presence and absence of Ca²⁺ using single-particle cryo-electron microscopy (cryo-EM), we reveal Ca²⁺-dependent conformational changes in the transmembrane domain and β-linkers. Electrophysiological experiments further show that a glutamate residue in the extracellular vestibule of the pore underpins the Ca²⁺-binding site, whose occupancy determines the conformation of the pore and therefore ion flow through the channel. These results reveal the molecular principles governing gating of BASIC and its regulation by Ca²⁺ ions, demonstrating that Ca²⁺ ions modulate BASIC function via changes in protein conformation rather than solely from a pore-block, as proposed for other members of this superfamily.

In mammals, the ENaC/DEG superfamily of voltage-insensitive, Na⁺-selective ion channels comprises three distinct subfamilies: BASIC, proton-gated acid-sensitive ion channels (ASICs), and constitutively active epithelial Na⁺ channels (ENaCs)[1,2]. Despite being the least characterized member[3,4], BASIC is expressed in both the nervous system and epithelial tissues[3,5], in contrast to the more restricted expression of ASICs in the nervous system and ENaCs in the kidney, lungs, and colon. Indeed, recent studies have shown that BASIC is important for maintaining the intrinsic excitability of the cerebellum[6] as well as regulating ion transport in the liver[4]. Additionally, a missense mutation of BASIC is associated with recurrent pregnancy loss in humans, suggesting the channel is essential for embryonic survival[7]. However, the mechanism underlying BASIC activation is poorly understood. High millimolar concentrations of bile acid are thought to activate BASIC in the liver and intestines[4,8–12] but the stimulus that activates cerebellar BASIC remains unknown. $Ca^{2+}_o$, which causes channel quiescence at physiological concentrations (-1.3 mM), promotes robust, non-desensitizing inward Na⁺ currents when removed[8,10,12]. Indeed, in the absence of any activator, a reduction in $Ca^{2+}_o$ leads to an increase in BASIC activity. Thus, although global levels of brain $Ca^{2+}_o$ are tightly regulated within a narrow range, local $Ca^{2+}_o$ fluctuations represent a situation in which BASIC channels might be activated[13–17]. In the cerebral cortex, this might occur when local Ca²⁺ homeostasis is transiently attenuated, such as during slow sleep oscillations, high neuronal activity, seizures, and ischemia[18–26].

Sensitivity to $Ca^{2+}_o$ appears to be a salient characteristic of the ASIC subfamily, particularly ASIC3[27–32]. Although ASIC3 is activated by low pH, it can also be activated at physiological pH during local reductions in $Ca^{2+}_o$ resulting from lactate-dependent Ca²⁺ chelation[33]. Functional studies on rat ASIC3 (rASIC3) have suggested that Ca²⁺ occupies a high-affinity Ca²⁺-binding site within the pore, preventing Na⁺ from permeating. In this pore-block model[30], channels become active upon relief of Ca²⁺ block due to either an increase in the concentration of protons, which compete for the Ca²⁺-binding site, or a

[1]Vollum Institute, Oregon Health and Science University, 3232 SW Research Drive, Portland, OR, USA. [2]Howard Hughes Medical Institute, Oregon Health and Science University, 3232 SW Research Drive, Portland, OR, USA. ✉e-mail: gouauxe@ohsu.edu

reduction in $[Ca^{2+}]_o$ at physiological pH. Release of bound $Ca^{2+}$ from the pore triggers $Na^+$ permeation without necessitating conformational alterations in channel structure. Subsequent investigations identified a putative $Ca^{2+}$-binding site, formed by residue E435, located in the region of the pore[34,35]. Interestingly, this residue is conserved in BASIC (E440) but absent in ASIC family members with lower $Ca^{2+}$ sensitivity. We therefore hypothesized that BASIC channels are gated open by depletion of $Ca^{2+}_o$.

An architectural model is required to understand the molecular principles of BASIC gating, but no structural information is currently available for BASIC. There is, however, a conserved architecture among other members of the ENaC/DEG superfamily[36]. These trimeric channels resemble a chalice in which a large extracellular domain (ECD) connects to a slender transmembrane domain (TMD) via linkers at the membrane interface. In this study, we used single-particle cryo-EM to determine the structure of hBASIC in the membrane-like environment of lipid nanodiscs, confirming its global adherence to the canonical ASIC fold. By determining structures in the presence and absence of $Ca^{2+}$, we identified a single $Ca^{2+}$-bound, non-conducting conformation of hBASIC, as well as both conducting and non-conducting conformations of $Ca^{2+}$-free BASIC, revealing unexpected $Ca^{2+}$-dependent conformational changes. Furthermore, using electrophysiology, we identified a preference for $Ca^{2+}$ over other divalent ions and a voltage sensitivity associated with $Ca^{2+}$ binding. Our study provides a structure-based mechanism to describe the interplay between $Ca^{2+}_o$ and channel gating in hBASIC and, by extension, the ASIC family.

## Results

### hBASIC adopts a modified ASIC fold

To verify that hBASIC is modulated by $Ca^{2+}$, we injected *Xenopus* oocytes with mRNA encoding full-length hBASIC and recorded current change in response to the removal of $Ca^{2+}_o$ using two-electrode voltage clamp (TEVC). $Ca^{2+}$ chelation via ethylene glycol-bis (β-aminoethyl ether)-N,N,N',N'-tetraacetic acid (EGTA) resulted in a sustained inward $Na^+$ current (Supplementary Fig. 1a–c), which was attenuated by the hBASIC pore blocker diminazene (Supplementary Fig. 1d) and potentiated by the bile acid deoxycholic acid (Supplementary Fig. 1e)[4,10,12,37]. The half-maximal inhibitory concentration ($IC_{50}$) of $Ca^{2+}$ ($7 \pm 1 \mu M$) (Fig. 1a) was similar to previously reported values[10,12].

We sought to reconstitute hBASIC in a membrane-like environment to capture physiologically relevant structures of hBASIC. Because chicken ASIC1a (cASIC1a) has a disordered lower pore in structures of detergent-solubilized protein, we considered extracting BASIC from membranes using the polymer styrene-maleic acid (SMA), which resulted in well-resolved cASIC1a pores[38]. However, as the available SMA copolymers chelate divalent ions, we instead solubilized hBASIC using the sterol-based detergent digitonin before reconstitution in lipid nanodiscs. To verify that digitonin extraction of hBASIC did not irreversibly compromise the ion channel activity of hBASIC, we expressed and purified full-length hBASIC using the protocol used for our structural studies (Supplementary Fig. 2a–c) and reconstituted the protein in proteoliposomes for injection into oocytes. Upon chelation of $Ca^{2+}_o$, oocytes injected with purified hBASIC exhibited robust inward $Na^+$ currents that were attenuated by diminazene (Supplementary Fig. 2d–f), confirming the integrity of hBASIC following our purification and reconstitution procedures.

To investigate the structure of hBASIC in its resting state, we performed single-particle cryo-EM analysis of hBASIC reconstituted in nanodiscs in the presence of 2 mM $Ca^{2+}_o$ (Supplementary Fig. 2g–i), revealing a single conformation with a global resolution of 2.89 Å (Supplementary Fig. 3). The resulting density map enabled residues 29-479 of hBASIC to be fitted, with unambiguous assignment of most side chains (Supplementary Fig. 4, and Table 1). Five *N*-linked glycan molecules per protomer were assigned at sites predicted from sequence analysis and deglycosylation assays confirmed heavy

glycosylation of hBASIC (Supplementary Fig. 2c, i). Consistent with the structures of cASIC1a and the invertebrate FMRFamide-gated $Na^+$ channel (FaNaC1/FMRFa), the carboxy-terminus (residues 479-505) and parts of the amino-terminus (residues 1-28) were not resolved[38,39], presumably due to disorder. Prior single channel recordings of hBASIC have revealed that it predominantly exists in a non-conducting state in the presence of high mM $Ca^{2+}_o$[12]. We therefore propose that the sole conformation observed in our 2 mM $Ca^{2+}$ data set represents the channel in a non-conducting state and refer to this structural model as '$Ca^{2+}$-closed'.

The overall architecture of hBASIC in the presence of $Ca^{2+}$ adheres to the canonical ASIC fold, a conserved architecture among the ENaC/DEG superfamily of trimeric ion channels[36]. Alignment of the main chain atoms with cASIC1a[38] reveals an α-carbon root mean square deviation (rmsd) of 1.5 Å and 1.8 Å for the desensitized and resting states, respectively, indicative of high structural conservation (Supplementary Fig. 5a). Indeed, each protomer resembles the characteristic upright clenched hand holding a ball, with the five domains of the ECD (palm, thumb, finger, knuckle, and β-ball) connected to the TMD via linkers at the membrane interface (wrist) (Fig. 1b). The amino-terminal transmembrane helix, TM1, forms a continuous α-helix of mostly hydrophobic residues that is poised to make direct contact with membrane lipids as it passes through the bilayer. A pre-TM1 re-entrant loop, containing the conserved 'HG' (His-Gly) motif, lines the bottom of the pore to create a constriction in the lower ion permeation pathway. In contrast, TM2 is a non-continuous α-helix comprising two segments, TM2a and TM2b, bridged by the 'GAS' (Gly-Ala-Ser) motif, which forms the central constriction site within the permeation pathway.

To ascertain whether sequence differences impact domain and secondary structure, we compared the protomer structure of $Ca^{2+}$-closed hBASIC against two non-conducting structures of cASIC1a; the low-pH-desensitized and high-pH-resting states (Supplementary Fig. 5b–c)[38]. The TMD and the scaffold of the ECD (palm and β-ball domains) show high structural conservation, with rmsds ranging between 0.4 and 2.6 Å. hBASIC deviates substantially from cASIC1a in the finger and thumb domains, exhibiting rmsds greater than 5 Å in these regions (Supplementary Fig. 5a). In hBASIC, the finger domain is a compacted helical bundle comprising three continuous helices connected by short loops. In contrast, the finger domain of cASIC1a contains long loops and multi-segmented helices. In the thumb domain, although both paralogs share placement of their α4 helix, the positioning of their α5 helices diverge (Supplementary Fig. 5b–c).

This structural diversity reflects fundamental functional differences between hBASIC and ASIC, which have distinct activation mechanisms and responses to $Ca^{2+}_o$. In particular, ASIC family members respond to extracellular acidification with rearrangements of the finger and thumb domains, such that low and high $[H^+]_o$ result in expanded and collapsed conformations, respectively[40]. In contrast, hBASIC is not gated by protons. Furthermore, whereas $Ca^{2+}_o$ strongly inhibits the activity of hBASIC, it merely affects the steady-state desensitization and pH-sensitivity of ASIC1a. Our structural data suggest these differences arise from distinct $Ca^{2+}$-binding sites. In ASIC1a, $Ca^{2+}$ binds within a cavity of negative electrostatic potential at the interface of the finger and thumb domains, known as the acidic pocket (Supplementary Fig. 5d–e)[27,32]. However, the carboxyl-carboxylate pairings that confer the $Ca^{2+}$-binding site in ASIC are not conserved in hBASIC (Supplementary Fig. 5f), and we found no evidence of $Ca^{2+}$ binding in the 2 mM $Ca^{2+}$ map at or near the interface of the finger and thumb domains.

### $Ca^{2+}$ removal promotes a conducting state of hBASIC

To reveal how hBASIC responds to $Ca^{2+}$ removal, we prepared grids of hBASIC reconstituted in nanodiscs in the presence of 2 mM EGTA to chelate ambient $Ca^{2+}_o$ and analyzed the sample using single-particle

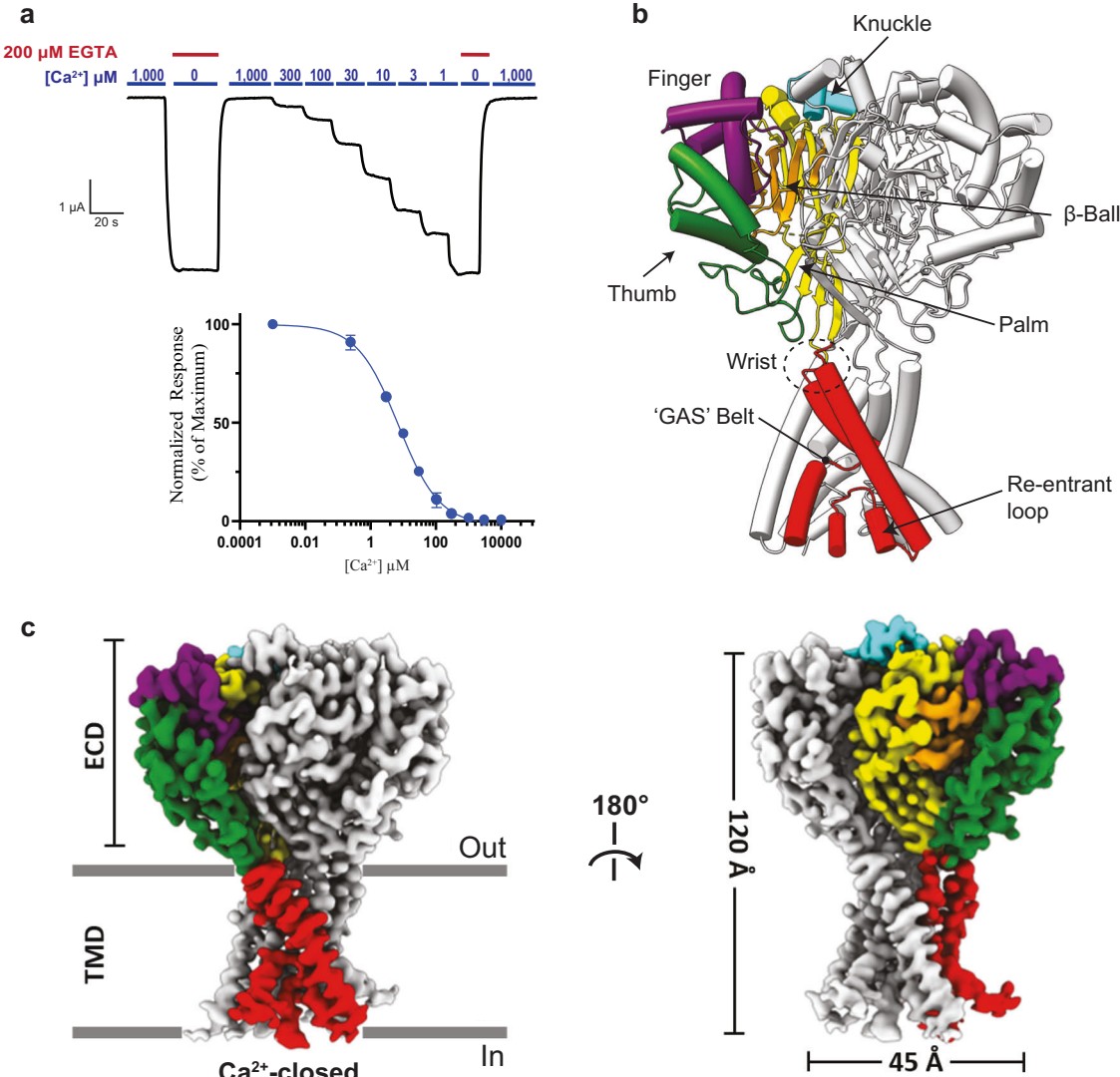

**Fig. 1 | Functional and structural characterization of hBASIC. a** $Ca^{2+}$ dose-response assay for hBASIC expressed in *Xenopus* oocytes. TEVC recordings in response to decreasing $Ca^{2+}$ concentrations (blue) and $Ca^{2+}$ removal by perfusion of EGTA (red). $IC_{50}$ for $Ca^{2+}$ = 7 μM ± 1 μM. Error bars represent SEM, center represents the mean, $n = 6$, representing recordings from 6 distinct oocytes. Source data are provided as a Source Data file. **b** Structure of hBASIC. A single subunit colored by domain organization reveals a similar fold to ASIC1a. **c** Side view of electron potential map of hBASIC in a nanodisc in the presence of $Ca^{2+}$. A single protomer is colored according to domain organization in (**b**).

cryo-EM. The initial 3D reconstruction of hBASIC was anisotropic and incomplete, containing a high-resolution ECD but an unresolved TMD. Following extensive 3D classification methods, two predominant classes emerged, each containing between 15,000-20,000 particles. The overall resolution of both classes was lower than hBASIC in the presence of $Ca^{2+}$, with both class 1 and class 2 having a global resolution of ~3.4 Å (Supplementary Fig. 6). Nevertheless, both density maps were sufficiently resolved to achieve placement of the TMD backbone with the assistance of several distinct hydrophobic side chains and models of hBASIC with $Ca^{2+}$ and cASIC1a (Supplementary Figs. 7, 8, and Table 1). Single channel recordings show that hBASIC transitions between open and closed states in the absence of $Ca^{2+}_o$[12]. In accordance with the electrophysiology data, we observed two distinct particle classes that differ in their pore conformations, which we thus refer to as 'EGTA-closed' (class 1), representing a non-conducting state, and 'EGTA-open' (class 2), representing an expanded, conducting state.

By comparing the $Ca^{2+}$-closed and EGTA-open structures of hBASIC, we gained mechanistic insight into how $Ca^{2+}$ influences channel gating. Structural superposition revealed clear conformational

rearrangements associated with $Ca^{2+}$ removal, which was surprising for two reasons (Fig. 2a). First, the pore-block model for rASIC3 predicts that removal of $Ca^{2+}$ eliminates a plug in the ion channel pore without the need for conformational changes[30]. Second, the largest conformational changes in hBASIC were at the TMD and the β-linkers (Fig. 2b–d, and Supplementary Figs. 9, 10), indicating that rearrangements driving channel activation originate in the TMD. This is in sharp contrast to the established proton-dependent mechanism of ASIC activation, in which major conformational shifts in the finger and thumb domains, far from the TMD, drive channel activation[40].

The overall architecture of the TMD is conserved in the three structural states of hBASIC, adopting a domain-swapped conformation of TM2a and TM2b (Fig. 2d, and Supplementary Fig. 9a). This is in contrast to the heterogeneity observed in conducting structures of cASIC1a—domain swapping of TM2 helices being evident in the MitTx-cASIC1a complex[41] but not in the PcTx1-cASIC1a complex[42]. In hBASIC, $Ca^{2+}$ removal induces a rotation of TM1 and TM2a away from the three-fold axis, resulting in a 4 Å helical displacement of TM1 and a 6° rotation of TM2a (Fig. 2d, and Supplementary Fig. 9). Although the

**Table 1 | cryoEM data collection, processing and validation statistics**

| | Ca²⁺ - Closed EMBD-48342 PDB: 9MKY | EGTA, Closed EMBD-48380 PDB: 9MLV | EGTA, Open EMBD-48343 PDB: 9MKZ | Ba²⁺ - Closed EMBD-48570 |
|---|---|---|---|---|
| **Data collection/processing** | | | | |
| Microscope | Krios (PNCC) | Krios (PNCC) | Krios (PNCC) | Krios, (HHMI) |
| Camera | K3 | Falcon 4i | Falcon 4i | Falcon 4i |
| Magnification (kX) | 165 | 165 | 165 | 165 |
| Voltage (kV) | 300 | 300 | 300 | 300 |
| Frames (no.) | 50 | 1096 | 1096 | 1096 |
| Electron exposure (e⁻/Å²) | 50 | 50 | 50 | 50 |
| Defocus range (um) | −0.8 to −2.4 | −0.8 to −2.4 | −0.8 to −2.4 | −0.8 to −2.4 |
| Pixel size (Å) | 0.41275 | 0.814 | 0.814 | 0.743 |
| Initial micrographs (no.) | 15,129 | 13,174 | 13,174 | 6017 |
| Micrographs used (no.) | 12,832 | 10,763 | 10,763 | 4988 |
| Particles picked (no.) | 6,526,929 | 5,741,395 | 5,741,395 | 2,425,855 |
| 3D-cleaned particles (no.) | 261,086 | 178,773 | 178,773 | 114,415 |
| **Final Refinement** | | | | |
| Symmetry imposed | C3 | C3 | C3 | C3 |
| Final particle (no.) | 87,709 | 15,003 | 18,221 | 19,511 |
| Map Resolution (A) | 2.89 | 3.43 | 3.47 | 3.04 |
| FSC threshold | 0.143 | 0.143 | 0.143 | 0.143 |
| **Model Statistics** | | | | |
| Initial model used | Alphafold | PDB: 9MKY | PDB: 9MKY | |
| Model resolution (A) | 3.1 | 3.8 | 3.9 | |
| FSEC threshold | 0.5 | 0.5 | 0.5 | |
| Map sharpening B factor (A) | N/A | N/A | N/A | |
| **Model composition** | | | | |
| Non-hydrogen atoms | 10146 | 9705 | 9576 | |
| Protein residues | 1266 | 1272 | 1245 | |
| Ligands | 15 | 0 | 0 | |
| **B factors (Å²)** | | | | |
| Protein | 51.24 | 92.64 | 47.28 | |
| Ligand | 128.33 | 0 | 0 | |
| **r.m.s.deviations** | | | | |
| Bond lengths (Å) | 0.004 | 0.003 | 0.003 | |
| Bond Angles (o) | 0.523 | 0.544 | 0.611 | |
| **Validation** | | | | |
| MolProbity score | 1.27 | 1.38 | 1.37 | |
| Clashscore | 3.88 | 4.12 | 3.96 | |
| Poor Rotamers (%) | 0.19 | 0 | 0 | |
| **Ramachandran plot** | | | | |
| Favored (%) | 97.53 | 96.9 | 96.92 | |
| Allowed (%) | 2.47 | 3.1 | 3.08 | |
| Disallowed (%) | 0 | 0 | 0 | |

resolution of our maps does not allow definitive placement of side chains, the observed backbone rotation suggests reduced intersubunit distances, which could favor stabilization of intermediate states between conducting and non-conducting conformations. Indeed, a salt bridge interaction between E440 in TM2a, and R83 in the adjacent TM1, bears a striking resemblance to that in a predicted intermediate state of rASIC3. Atomistic molecular dynamics simulations of rASIC3 carrying the G429E mutation using the cASIC1 model revealed an expanded, but non-conductive, intermediate state between the Ca²⁺-blocked and open conformations in which an interaction was formed between equivalent residues (E426 and R61)[35]. The congruence of our structural data with these molecular dynamics simulations suggests

that this arginine, which is conserved throughout the superfamily, stabilizes gating transitions between Ca²⁺-bound and -unbound states.

Two short regions of polypeptide linking the upper and lower palm domains, β11-β12 and β1-β2, have long been established as critical components of gating in ASIC channels[43–46]. β11-β12, known as the 'molecular clutch' of the channel, acts as the structural link between the ECD and the TMD[40] (Fig. 2c, and Supplementary Fig. 9a, b,10a, b). In cASIC1a, reorientation of β11-β12 during desensitization decouples the ECD from the TMD, facilitating conformational changes of the pore without necessitating movement of the ECD. Interestingly, the β11-β12 linker in our Ca²⁺-closed hBASIC structure bears a striking resemblance to that in the desensitized conformation of cASIC1a, notably, the

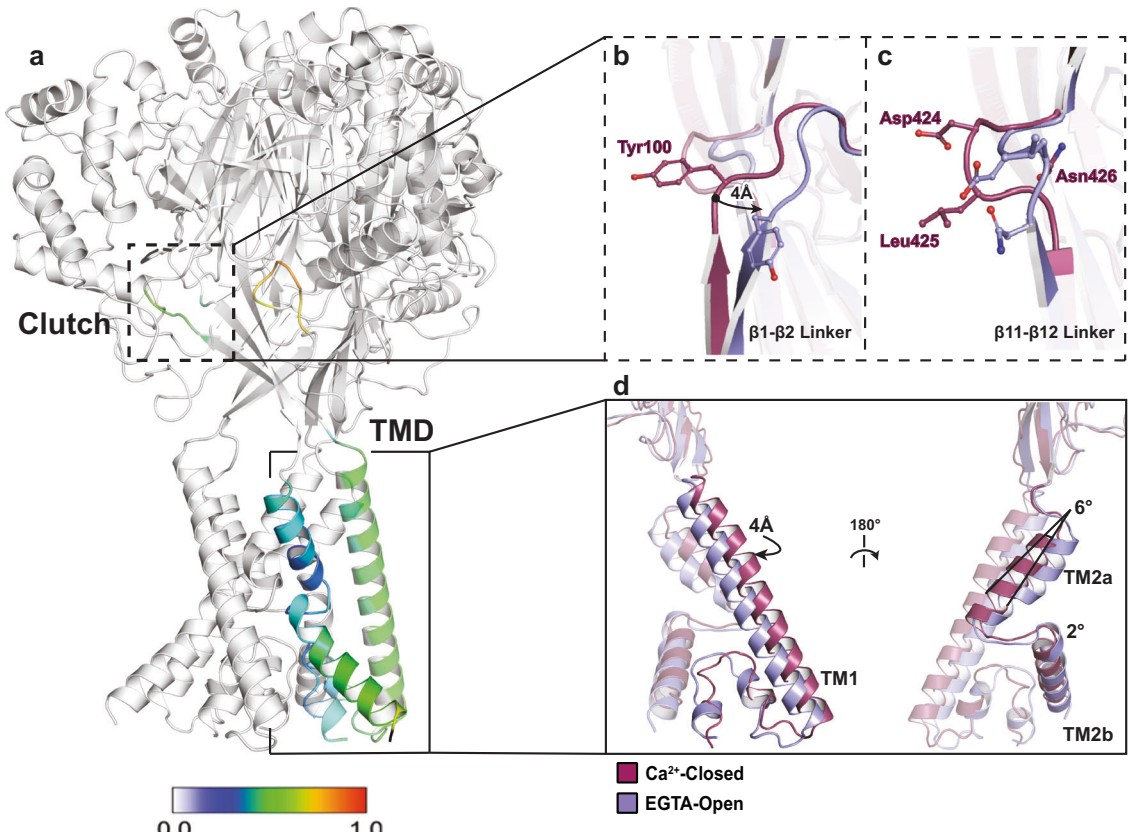

**Fig. 2 | Ca²⁺-dependent conformational changes of hBASIC. a** hBASIC trimer assembly. A single-subunit of the hBASIC Ca²⁺-closed model colored according to displacement of backbone α-carbon atoms in comparison to the EGTA-open model. Clutch and TMD regions show greatest level of displacement between the two models. **b, c** Superposition of Ca²⁺-closed (burgundy) and EGTA-open (lavender) states at the β1-β2 (**b**) and β11-β12 (**c**) linkers. **d** Single subunit superposition of the TMD highlights changes in TM1, TM2a, and TM2b.

conserved residues L425 and N426. L425 is oriented towards the central vestibule and N426 forms a hydrogen bond with D424 and the backbone of V101 in the β1-β2 linker (Fig. 2c, and Supplementary Fig. 10c, d). Removal of Ca²⁺ promotes conformational heterogeneity in this molecular clutch. EGTA-closed resembles the Ca²⁺-closed conformation whereas EGTA-open is dramatically reoriented such that the positions of the L425 and N426 side chains are switched; N426 now facing the vestibule and L425 making contacts with the β1-β2 linker (Fig. 2c, and Supplementary Fig. 10e-f). This is reminiscent of the conformation seen in the MitTx-bound open structure of cASIC1a[41].

Similarly, the β1-β2 linker, a sequence-diverse region in ASICs, thought to modulate gating via its interaction with the molecular clutch[47,48], has a distinct conformation in Ca²⁺-closed and EGTA-open hBASIC (Fig. 2b). In the Ca²⁺-closed structure, the β1-β2 linker hovers just over the β11-β12 linker, with the backbone placement resembling that in the high pH structure of cASIC1a at rest (Fig. 2b, and Supplementary Fig. 10a–d). Unique to hBASIC is a tyrosine residue (Y100) that inserts itself towards the central vestibule, adjacent to L425 (Fig. 2b, and Supplementary Fig. 10b). Its aromatic ring sterically blocks the outward movement of the β11-β12 linker, preventing L425 from exchanging positions with N426 (Fig. 2c, and Supplementary Fig. 10g). Removal of Ca²⁺ triggers a 4 Å lateral displacement of the β1-β2 linker, repositioning the tyrosine residue and promoting rearrangement of the β11-β12 linker to open the pore (Supplementary Fig. 10h). Thus, the β1-β2 linker stabilizes the closed conformation by reducing the capacity for movement into the open state.

## Ca²⁺ removal drives expansion of the pore

In the ASIC family, ions enter the pore at the extracellular vestibule, a V-shaped fenestration that starts at the junction between the ECD and

TMD and extends 5-10 Å into the lipid bilayer (Fig. 3a, and Supplementary Fig. 11a-c). Enriched with acidic residues, the vestibule is thought to concentrate cations near the entrance of the pore[49], a feature that appears to be conserved in hBASIC. In the Ca²⁺-closed structure, E440 and D444 are orientated upwards, forming two triangular antiprisms of negative electrostatic potential. Removal of Ca²⁺ leads to rotation of the TM2 helix, displacing these residues from the center of the pore and expanding the extracellular vestibule (Fig. 3b, and Supplementary Fig. 11a-c). This results in antiprism widening from 12 Å to 15 Å at E440 Cα and from 7 Å to 12 Å at D444 Cα between the Ca²⁺-closed and EGTA-open state (Fig. 3c, and Supplementary Fig. 11d-e).

Below the triangular rings of charged residues lies a major constriction of the permeation pathway formed by a ring of carbonyl oxygen atoms from G447 (Fig. 3d, and Supplementary Fig. 11d-e). In both the Ca²⁺-closed and EGTA-closed structures, this 'glycine gate' is impermeable to ions, with pore radii less than 1 Å and ~1.2 Å, respectively (Fig. 3b, and Supplementary Fig. 11a–c). The gate is significantly widened in the EGTA-open structure, having a pore radius greater than 2 Å, which is sufficient to accommodate partially dehydrated Na⁺ ions (Fig. 3b, and Supplementary Fig. 11c). A permeating cation would subsequently meet the GAS motif before continuing to the final constriction in the permeation pathway formed by the re-entrant loop. The HG motif of the re-entrant loop points directly into the pore in the Ca²⁺-closed structure, obstructing the pathway of ions. Removal of Ca²⁺ causes rearrangement of this loop to displace the histidine and increase the pore-radius from ~1 Å to ~3 Å (Fig. 3b). These findings are consistent with subtle changes in the conformation of the re-entrant loop in different structural states of FaNaC1/FMRFa[39], providing additional support for the role of the re-entrant loop in gating and ion conduction.

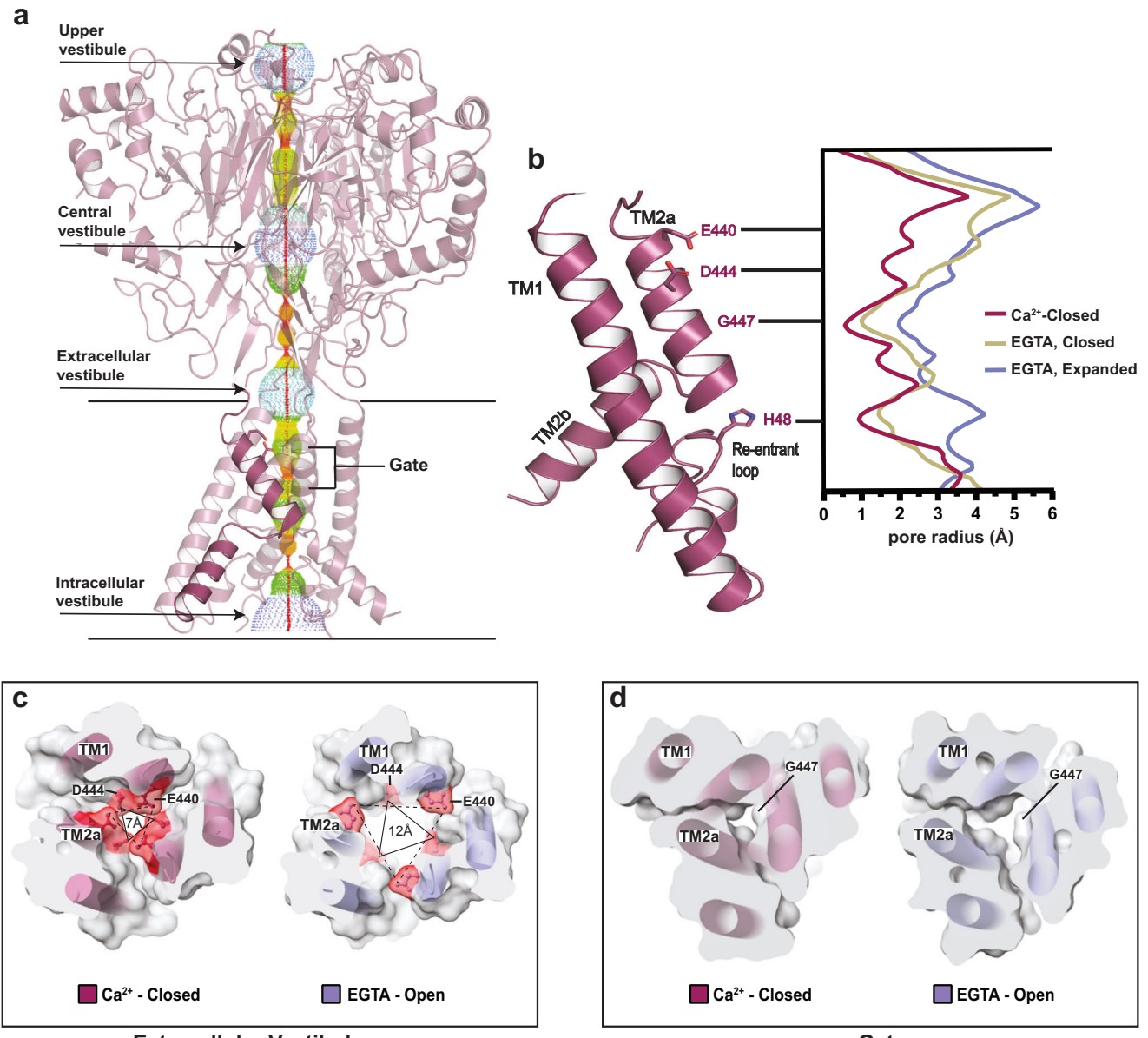

**Fig. 3 | State dependent pore conformation. a** Ca²⁺-closed pore profile calculated with HOLE software. Pore radii: red <1.15 Å, green <2.3 Å, <purple. **b** Plot of pore radius for Ca²⁺-closed (burgundy), EGTA-open (lavender), and EGTA-closed (tan) channels. Source data are provided as a Source Data file. **c, d** Top-down view of cross-sectional slices of the Ca²⁺-closed and EGTA-open pores at the extracellular vestibule (**c**) and gate (**d**). Key residues, E440, D444, and G447, highlighted to show changes in distance in response to Ca²⁺ removal. Triangles show distance increase at residue D444.

## E440 forms a binding site for hydrated Ca²⁺

Analysis of the Ca²⁺-closed hBASIC map revealed a cylindrical Ca²⁺ ion-like density at the apex of the channel pore, a feature that was absent in both EGTA maps. Among the residues within interacting distance of this density, E440 is unique to the Ca²⁺-sensitive channels in this superfamily (Fig. 4a, b). Indeed, in rASIC3, the corresponding residue is pivotal for Ca²⁺ sensitivity[35]. By contrast, in family members that exhibit weak Ca²⁺ sensitivity, such as ASIC1a, the corresponding residue is a glycine or alanine (Fig. 4b). We observed ~3 Å displacements of the E440 Cα between the EGTA-closed and EGTA-open conformations, relative to the Ca²⁺-open structure (Fig. 4c). We thus hypothesized that the three additional carboxylate side chains contributed by E440 in hBASIC enhance its Ca²⁺ sensitivity. To test this, we generated an E440Q mutant, which reduced Ca²⁺ sensitivity by approximately 10-fold relative to wild-type (WT), and an E440G mutation to mimic ASIC1a, which resulted in an approximate 100-fold reduction in

potency (Fig. 4d, Supplementary Fig. 12a-b). These site-directed mutants confirm that E440 forms a critical Ca²⁺-binding site at the apex of the pore.

Docking a Ca²⁺ ion into the ion-like density in the Ca²⁺-closed BASIC map revealed ~4.0 Å distances between the carboxyl oxygens of residue E440 and the putative ion (Fig. 4b). Given that the ionic radius of Ca²⁺ is ~1 Å, this distance is too large for direct coordination of a dehydrated ion, suggesting that instead, Ca²⁺ binds in a hydrated state. This geometry resembles the Ca²⁺-binding site at the entrance to the pore of voltage-gated Ca²⁺ channels, where interactions with Ca²⁺ occur through the inner hydration shell[50] and the fully hydrated ion is 4-5 Å away from the carboxyl oxygens. Considering the established role of water-mediated interactions in cation-binding sites, we suggest that E440 creates a malleable binding site in hBASIC, wherein protein ligands adapt their positioning according to the ion's physico-chemical properties[51].

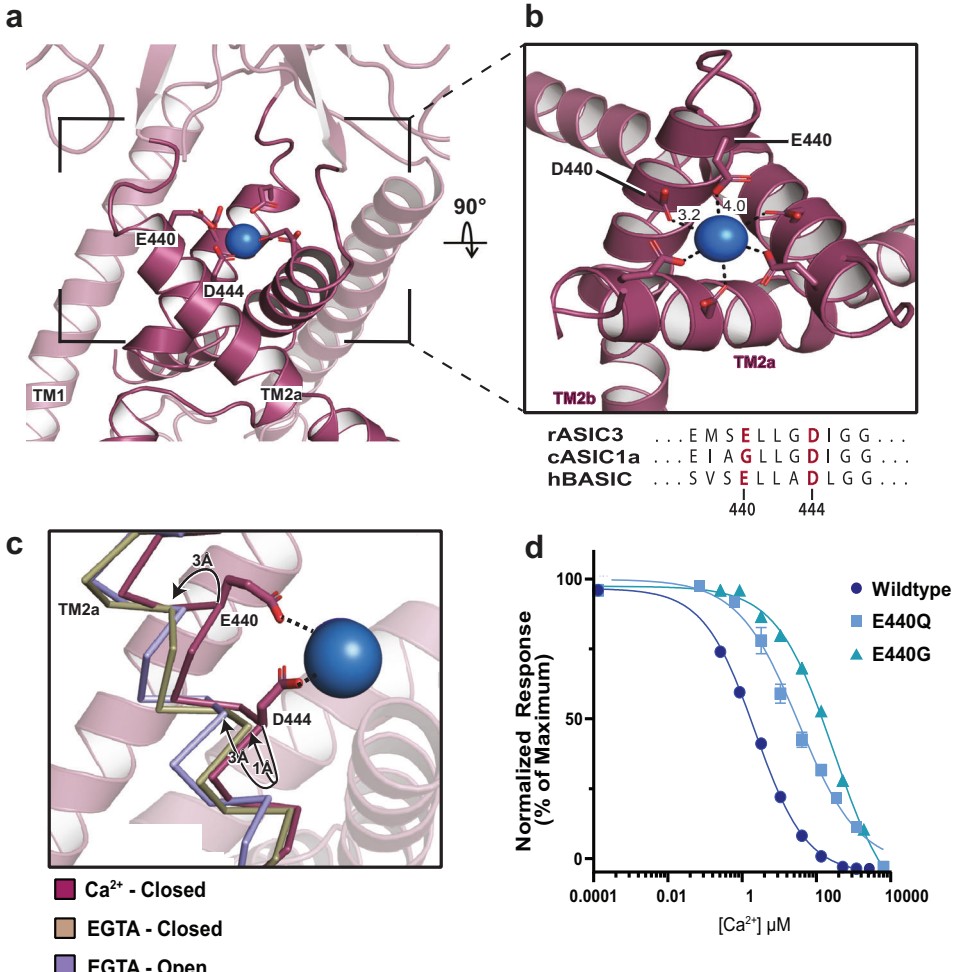

**Fig. 4 | The role of residue E440 in BASIC's $Ca^{2+}$ sensitivity. a** A $Ca^{2+}$ ion modeled into the density at the top of the channel pore. **b** Close up view of the $Ca^{2+}$ binding site, showing ion coordination by residues E440 and D444. $Ca^{2+}$-oxygen distances expressed in Å. **c** Superposition of TM2a in $Ca^{2+}$-closed (burgundy), EGTA-closed (tan), and EGTA-open (lavender) demonstrates that movement of backbone in response to removal of $Ca^{2+}$ displaces E440 and D444 away from $Ca^{2+}$. **d** $Ca^{2+}$ inhibition dose-response curve from TEVC recordings of oocytes expressing WT, E440Q, or E440G BASIC, resulting in $IC_{50}$ values of $7 \pm 1 \mu M$, $107 \pm 18 \mu M$, and $883 \pm 116 \mu M$, respectively. Error bars represent SEM, center represents the mean, $n = 6$, representing recordings from 6 distinct oocytes. Source data are provided as a Source Data file.

To assess the plasticity of the ion-binding site in hBASIC, we tested whether other divalent ions could occupy the site and effectively block the channel. Indeed, although previous studies have shown that hBASIC is sensitive to $Mg^{2+}$[12], it is unknown whether the binding site can accommodate ions of this size. We first evaluated the inhibition potency of $Mg^{2+}$, $Sr^{2+}$, and $Ba^{2+}$. Consistent with prior findings[12], $Mg^{2+}$ blocked the inward $Na^+$ current with an $IC_{50}$ value of $32 \pm 2 \mu M$ (Fig. 5a, and Supplementary Fig. 12c-d). Moreover, despite their larger size, $Sr^{2+}$ and $Ba^{2+}$ also inhibited inward $Na^+$ currents ($IC_{50} = 290 \pm 29 \mu M$ and $57 \pm 4 \mu M$, respectively) (Fig. 5a, and Supplementary Fig. 12c, d). Thus, although the binding site favors $Ca^{2+}$ over other divalent ions, the ability of larger cations to inhibit the channel confirms that the site is flexible enough to accommodate ions of various size. If ion size were the only factor influencing binding, we would expect binding preference to follow the order $Mg^{2+} > Ca^{2+} > Sr^{2+} > Ba^{2+}$, contrary to our observations. The preferential order of ions binding in a hydrated state, however, would depend on additional factors such as hydrated radius, number of water molecules, and lifetime exchange rates[52]. Our results therefore support the notion that water molecules are key to the $Ca^{2+}$-binding site being able to accommodate a range of divalent ions.

The ability of $Ba^{2+}$ to block hBASIC presented an opportunity to further validate the identity of the non-protein density at the top of the

pore in the $Ca^{2+}$-closed map. If this is the site of $Ca^{2+}$ binding, we would expect a $Ba^{2+}$-bound map to show density within interacting distance of E440, given that a range of divalent ions can bind to and inhibit the channel. We also anticipated a stronger density signal for $Ba^{2+}$, due to its higher electron count and the dependence of electron scattering on atomic number. To test this, we captured hBASIC in nanodiscs in the presence of $Ba^{2+}$ ($Ba^{2+}$-closed) using single-particle cryo-EM (Supplementary Fig. 13, and Table 1). As expected, we observed a similar non-protein density at the top of the channel pore, within binding distance of E440. Superposition of the $Ba^{2+}$-closed and $Ca^{2+}$-closed maps revealed overlapping densities, with $Ba^{2+}$ exhibiting a stronger signal than $Ca^{2+}$ (Fig. 5b, and Supplementary Fig. 14). Together, these findings provide strong evidence that E440 plays a crucial role in forming a $Ca^{2+}$-binding site at the apex of the pore that broadly interacts with divalent ions.

## $Ca^{2+}$ block is voltage- but not pH-dependent

The location of the $Ca^{2+}$-binding site within the pore suggests that $Ca^{2+}$ inhibits hBASIC by acting as a physical barrier to ion flow. Further, being positioned within the permeation pathway implies that bound $Ca^{2+}$ would likely sense and be affected by the membrane's electric field. To address whether $Ca^{2+}$ block of hBASIC is sensitive to voltage

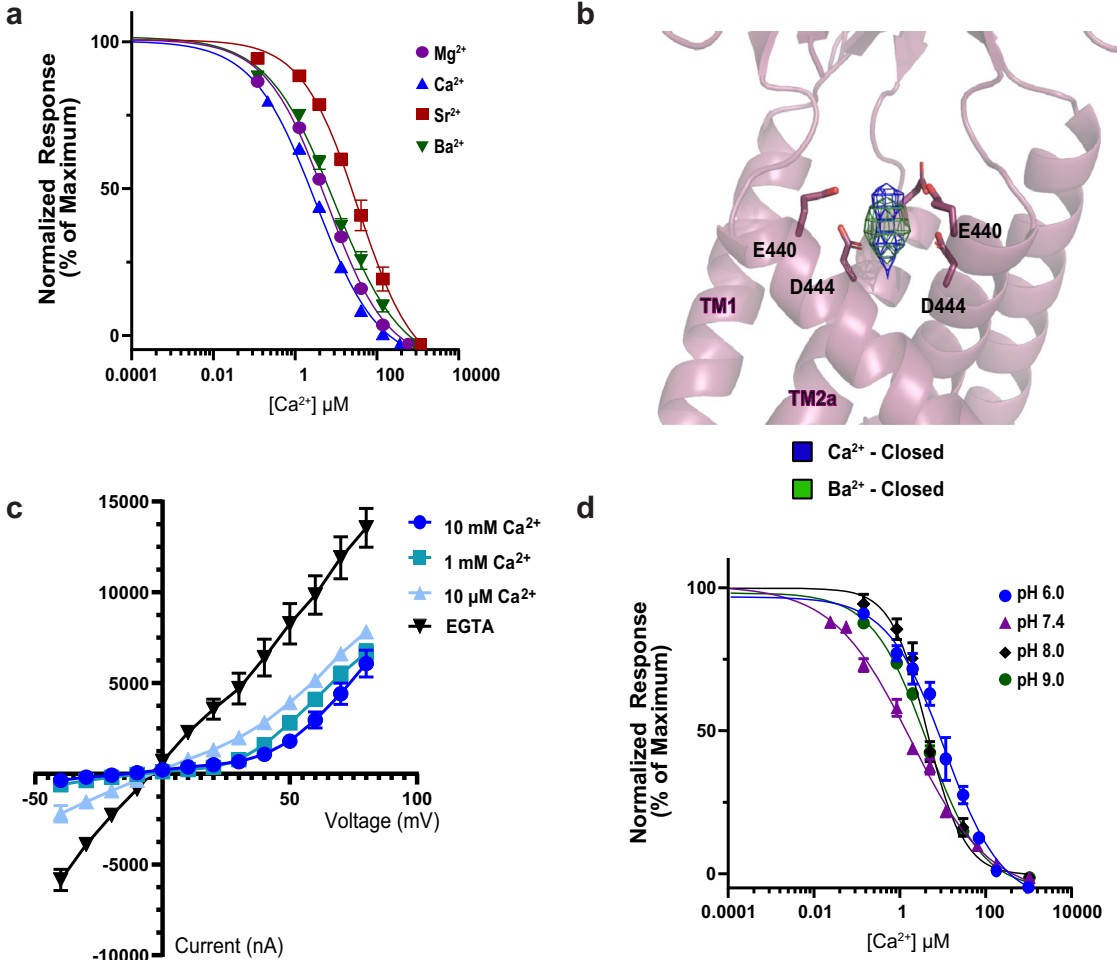

**Fig. 5 | Biophysical analysis of the Ca²⁺ binding site at the pore's apex.**
**a** Inhibition dose-response curves for $Mg^{2+}$, $Sr^{2+}$, $Ca^{2+}$, and $Ba^{2+}$ from TEVC recordings of oocytes expressing BASIC. **b** Superpositions of the ion-like densities at the top of the pore in the $Ca^{2+}$-closed and $Ba^{2+}$-closed cryo-EM maps. Isomesh map features are contoured at 5.0 σ for $Ca^{2+}$-closed and 6.0 σ for $Ba^{2+}$-closed, and within 2.5 Å associated with ion placement. **c** Current-voltage relationships from TEVC

recordings of oocytes expressing BASIC in the presence of EGTA, 10 μM, 1 mM, or 10 mM $Ca^{2+}$. **d** Inhibition dose-response curves for $Ca^{2+}$ in external recording solutions of different pH, resulting in IC₅₀ values of $31 \pm 2$ μM at pH 6.0, $7 \pm 1$ μM at pH 7.4, $9 \pm 1$ μM at pH 8.0, and $9 \pm 1$ μM at pH 9.0. Error bars represent SEM, center represents the mean, $n = 6$, representing recordings from 6 distinct oocytes. Source data are provided as a Source Data file.

we determined current-voltage relations in the presence and absence of $Ca^{2+}_o$ using TEVC of oocytes injected with hBASIC mRNA (Fig. 5c). In the absence of $Ca^{2+}$, hBASIC exhibited symmetrical inward and outward $Na^+$ currents, confirming the intrinsic voltage-insensitivity of the channel. Upon addition of $Ca^{2+}$ to the bath, currents were attenuated at negative potentials, resulting in an outwardly rectified current-voltage relation. The relief of block at positive potentials demonstrates that the action of $Ca^{2+}$ on the gating of hBASIC is voltage dependent, consistent with our structural data showing that there is a $Ca^{2+}$ binding site within the ion channel pore.

Finally, we considered whether $Ca^{2+}$ binding might be sensitive to changes in extracellular pH, given that the pKa of the carboxyl groups in the binding site could be influenced by local proton concentrations[30]. To assess this, we measured the effect of $Ca^{2+}$ on hBASIC currents by constructing dose-response curves at different pH values using TEVC. Lowering extracellular proton concentration to pH 8.0 or pH 9.0 resulted in comparable IC₅₀ values for $Ca^{2+}$ to that obtained at pH 7.4 ($7 \pm 1$ μM). At elevated proton concentrations, we observed a modest reduction in $Ca^{2+}$ block to an IC₅₀ of $31 \pm 9$ μM at pH 6.0 (Fig. 5d, and Supplementary Fig. 12e−f). These results suggest that the side chains of E440 remain relatively unaffected by pH within our testing range, consistent with predicted pKa values for both E440 and D444 falling below physiological pH fluctuations (pKa ~4.0). We thus

conclude that fluctuations in extracellular pH do not substantially influence $Ca^{2+}$-mediated inhibition of hBASIC.

## Discussion

We sought to define the structural principles governing $Ca^{2+}$ sensitivity in hBASIC. The interplay between $Ca^{2+}$ and gating has been a subject of investigation, not only for hBASIC, but also for the ASIC family. However, the overall mechanism has remained enigmatic due to a lack of structural information. Here, by obtaining cryo-EM structures of hBASIC in the presence and absence of $Ca^{2+}$, we have developed a structure-based model that describes how the removal of $Ca^{2+}$ results in channel gating.

The conserved aspartic acid ring directly above the gate (D444 in BASIC) has long been proposed as a key determinant of $Ca^{2+}$ sensitivity in BASIC[11], ASICs[28], and FaNaC/FMRFa[53]. Prior studies of cASIC1a crystals soaked in $Cs^+$ showed a similar elongated density in the same region of the extracellular vestibule as the cation-binding site in BASIC[49]. This elongated density comprised two $Cs^+$ ions, though likely not in simultaneously occupied binding sites, coordinated in a trigonal antiprism pattern by residues equivalent to D444 and the backbone of G443 in BASIC. The overlap in density shape and coordinating residue D444 provides evidence of a common cation-binding site among ASIC channels. However, the second carboxyl ring is unique to BASIC and

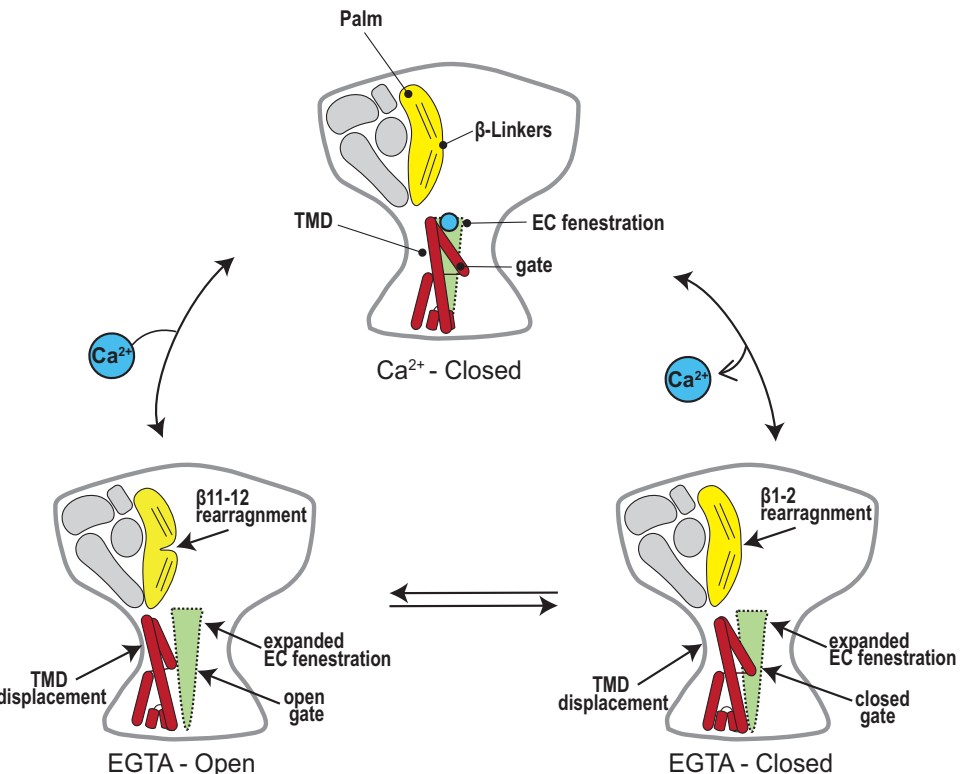

**Fig. 6 | Mechanism of Ca²⁺-dependent inhibition of hBASIC.** Ca²⁺ binds at the top of the channel pore, stabilizing the non-conducting state. A local decrease in Ca²⁺ₒ promotes Ca²⁺ dissociation, causing rearrangement of the TMDs, extracellular fenestrations, and β-linkers. Until Ca²⁺ is replenished, the channel stochastically cycles between a conducting and non-conducting state due to the opening and closing of the channel gate.

ASIC3; the equivalent residue being a glycine in cASIC1a. In agreement with prior studies of rASIC3, we show that the presence of a second carboxyl ring in BASIC enhances the inhibitory potential of Ca²⁺. Neutralizing this second charged ring of hBASIC with a cASIC1a-imitating E440G mutation significantly reduces the inhibitory potency of Ca²⁺, in accordance with previous estimates of low-affinity Ca²⁺ block of cASIC1a. It is thus likely that the mouth of the pore serves as a consensus Ca²⁺-binding site, with sensitivity to Ca²⁺ being dependent upon the number of carboxylic-acid-containing side chains.

Despite hBASIC and rASIC3 appearing to have an overlapping Ca²⁺-binding site, there are differences between the two. For example, we found that Ca²⁺ inhibition of hBASIC is not proton dependent, but the IC₅₀ for Ca²⁺ varies with pH in rASIC3, being more potent in alkaline conditions[30]. Secondly, Ca²⁺ block of hBASIC is influenced by membrane voltage whereas block of rASIC3 is insensitive to voltage[30]. As TEVC measurements are not direct assays of Ca²⁺ binding, it is likely that this difference is due to rASIC3 having multiple Ca²⁺-binding sites: one in the ECD that modifies sensitivity to protons and a second overlapping with that in hBASIC, which blocks the pore of the channel. Indeed, a recent study identified a Ca²⁺-binding site in the acidic pocket of rASIC3 that influences channel gating by modulating steady-state desensitization, in addition to the site in the pore[34]. Multiple binding sites would also explain why Ca²⁺ removal at pH 7.0 gives rise to desensitizing currents in rASIC3 but non-desensitizing currents at alkaline pH, similar to hBASIC. In accordance with this notion, there are at least two different gating mechanisms for rASIC3: (1) the canonical pH-dependent mechanism, in which protonation of an allosteric site in the acidic pocket induces conformational changes that result in desensitizing currents; and (2) a Ca²⁺-dependent mechanism, similar to that of hBASIC, in which release of Ca²⁺ from the top of the pore results in channel activation with persistent currents.

The structural and functional data presented here, together with findings from previous studies of BASIC and ASICs, illuminate the Ca²⁺-dependent gating observed in BASIC (Fig. 6). Under physiological Ca²⁺ concentrations, BASIC channels predominantly reside in a Ca²⁺-bound, non-conducting state, with brief excursions to an open-channel state. Ca²⁺ inhibits channel activity by directly binding to the carboxyl ring at the top of the pore, stabilizing the closed conformation of the TMD, which in turn locks the β1-β2 linker in a conformation that sterically prevents reorientation of the β11-β12 linkers that underlie channel activation. Transient and local decreases in Ca²⁺ₒ, such as during periods of high activity in the cerebellum, increase the likelihood of Ca²⁺ unbinding from the channel. This removal of Ca²⁺ triggers reorientation of the TMDs, resulting in the expansion of the extracellular fenestration. Displacement of TM1 causes relaxation of the β1-β2 linker toward the membrane, making room for the rearrangement of the β11-β12 linker, which results in dilation of the gate and ion flow through the pore. Until Ca²⁺ rebinds, BASIC stochastically transitions between conducting and non-conducting conformations as the gate opens and closes.

The Ca²⁺-dependent gating scheme we describe for BASIC expands our knowledge of activation mechanism for the broader ENaC/DEG family. Prior structural studies of cASIC1a have provided a detailed structural model explaining how proton-dependent conformation changes in the ECD, far away from the channel pore, can lead to channel activation[40]. Here, we show a distinct mechanism in which Ca²⁺-dependent conformational changes originating at the TMD prompt channel activation without necessitating conformational changes in the ECD. This represents gating in the ENaC/DEG family initiated from within the TMD. Central to both the BASIC and ASIC gating mechanisms are the β-linkers, underscoring their importance for channel gating. Acting as a molecular clutch, the β11-β12 linker is

associated with different orientations in closed and open conformations of the channel, whereas the ECD remains static. The β1-β2 linker stabilizes the closed state by hindering reorientation of the β11-β12 linker. An essential concept of the $Ca^{2+}$-dependent model could also explain how membrane-based substances, such as bile acid[4,54], fatty acids[55], and cholesterol[8], modulate the gating of BASIC and ASIC. Like $Ca^{2+}$, these membrane-active substances may influence channel gating by perturbing the conformation of the TMD, thereby indirectly modifying the β-linkers. Supporting this model, prior studies on rat BASIC demonstrates that a cytosolic α-helix regulates channel activity and mediates its sensitivity to cholesterol, with removal of the α-helix or depletion of cholesterol resulting in channel hyperactivity[8]. It is likely that the combined interplay between membrane-based substances and $Ca^{2+}$ serves to fine-tune channel activity in vivo. However, further research into interactions of membrane substances with BASIC and ASIC will be needed to uncover the mechanism of their membrane sensitivity.

## Methods

### Receptor construct
The cDNA encoding full-length wild-type human ASIC5 (NM_017419.3) was cloned into the pEG-BacMam vector for protein expression in human embryonic kidney cells (HEK293)[56]. To facilitate protein purification and analysis using fluorescent size-exclusion chromatography[57] (FSEC), the cDNA was engineered to include an 8x histidine tag, followed by an enhanced green fluorescent protein (eGFP) and a 3 C protease cleavage site at the N-terminal region of the ASIC5 gene. The resulting plasmid, designated as pEG-BacMam-Nterm-8xHis-eGFP-3C-ASIC5, was codon optimized for expression in mammalian cells and synthesized by Twist Biosciences.

### Protein expression and purification
For structural investigations, recombinant full-length BASIC was synthesized in HEK293T cells of female origin via baculovirus-mediated gene delivery. BASIC bacmids and baculoviruses were produced following established protocol[56]. HEK293T cells of female origin were cultured in Freestyle 293 Expression Medium (Gibco) supplemented with 2% fetal bovine serum (FBS) (Gibco) at 37 °C until reaching a density of $3.5 \times 10^6$ cells per milliliter before being inoculated with P3 virus. To enhance protein expression, the culture was supplemented with 10 mM sodium butyrate and 100 μM diminazene (Sigma), to final concentrations of 10 mM and 100 μM, respectively, before reducing the temperature to 30 °C six h post-infection. Following 24 h of expression, cells were harvested by centrifugation and washed in Tris-buffered saline containing $Ca^{2+}$ (150 mM NaCl, 20 mM Tris pH 8.0, 2 mM $CaCl_2$).

The cell pellets were resuspended in ice-cold Tris-buffered saline (TBS) supplemented with $Ca^{2+}$ and a cocktail of protease inhibitors (20 mM Tris pH 8.0, 150 mM NaCl, 2 mM $CaCl_2$, 1 mM phenylmethylsulfonyl fluoride, 0.05 mg/ml aprotinin, 2 μg/ml pepstatin A, and 2 μg/ml leupeptin). To facilitate solubilization, 2% digitonin in the same buffer was added to the suspension, achieving a final digitonin concentration of 1%. The cells were lysed at 4 °C with gentle stirring for 1 h, followed by ultracentrifugation for 45 minutes at 4 °C to remove insoluble debris. The resulting supernatant was incubated with GFP Nanobody (GNB) resin at 4 °C with gentle rocking for 1 h. The resin-bound protein complex was collected by centrifugation and transferred into a gravity column, where the resin was washed with 10 column volumes (CV) of TBS containing 0.1% digitonin. BASIC protein was eluted by overnight digestion with HRV-3C protease (Sigma) at 4 °C. The eluted protein was analyzed by SDS-PAGE and FSEC (excitation: 280 nm, emission: 330 nm), and the sample was concentrated to approximately 15 μM using a 100-kDa MWCO Amicon centrifugal filter (Merck Millipore).

### Nanodisc reconstitution
To reconstitute BASIC into nanodiscs, the purified protein was combined with lipids and MSP1D1 membrane scaffold protein at a molar ratio of 1:5:200 (trimer:MSP:lipids). The membrane scaffold protein was expressed and purified following previously established procedures without modifications[58]. Brain polar lipid extract (Avanti Polar Lipids) was prepared by dissolving the powder in chloroform and drying the mixture overnight under vacuum in a desiccator. The lipids were then resuspended in 20 mM Tris pH 8, 150 mM NaCl buffer to create a 20 mg/mL solution and subjected to iterative freeze-thaw cycles using liquid nitrogen and sonication until the solution was homogeneous. To prepare lipids for nanodisc reconstitution, the lipids were diluted to a 7.5 mg/mL solution using TBS supplemented with 40 mM n-dodecyl-β-D-maltoside (DDM) and 2.5 mM cholesteryl hemisuccinate (CHS). The solubilized lipids were incubated for 30 minutes with gentle agitation before being combined with purified hBASIC. Following a 30-minute incubation on ice, the membrane scaffold protein (MSP) was added.

For detergent removal to initiate reconstitution of BASIC into nanodiscs, equilibrated Bio-beads SM-2 resin (Bio-Rad) (500 mg/ml) was added to the sample. The Bio-beads underwent preparation by undergoing two washes with methanol, followed by rinsing with water, and finally with TBS. Bio-beads were replaced three times, each time undergoing a 2-h incubation at 4 °C with gentle agitation, with the final incubation lasting overnight at 4 °C with gentle agitation.

Following reconstitution, the sample was filtered to remove any aggregated material before purification via size-exclusion chromatography (SEC) on a Superose 6 increase 10/300 GL column (Cytiva) pre-equilibrated with TBS. SEC fractions were subjected to analysis by SDS-page and FSEC (excitation: 280 nm, emission: 330 nm). Peak fractions were collected and concentrated to approximately 3.5 mg ml$^{-1}$. To prevent preferred orientation, 100 μM of Fluorinated octyl maltoside (FOM) was added directly to the sample prior to adding either $Ca^{2+}$, $Ba^{2+}$, or EGTA, to a final concentration of 2 mM.

### EM sample preparation and data acquisition
Before applying the samples, Quantifoil 1.2/1.3 Au (300 mesh) grids were subjected to glow discharge (15 seconds at 15 mA) using the PELCO easiGlow system. Two aliquots of 3.5 μL BASIC protein were each applied to the grids in a 100% humidity environment at 4 °C. The first sample was manually blotted, while the second application was blotted with a blot force of 0 for 2.5 seconds before being rapidly plunged into liquid ethane using a Vitrobot Mark IV (Thermo Fisher). The grids were subsequently stored in liquid nitrogen until imaging.

Micrographs of nanodisc-embedded BASIC with $Ca^{2+}$ were collected at the Pacific Northwest Center for Cryo-EM (PNCC) on a 300 kEV Titan Krios transmission electron microscope (TEM) (Thermo Fisher) using a Gatan K3 direct electron detection camera. Data acquisition was automatic using serialEM[59] to identify holes using the hole finder features and combining for multi-shot multi-hole collection. 15,128 images were collected at a physical pixel size of 0.814 Å/pix (0.407 Å/pix with super-resolution) across a defocus range of −0.8 to −2.2 μm, with a total dose of 50 e⁻/Å$^2$ that was fractioned into 50 frames.

Micrographs of nanodisc-embedded BASIC with EGTA were collected at PNCC on a TEM (Thermo Fisher) using a Falcon4i camera and operating at 300 kV. A total of 13,174 movies were collected in electron event representation (EER) format with a physical pixel size of 0.814 Å/pix across a defocus range of −0.8 to −2.2 μm. A total of 6,017 cryo-EM images of nanodisc-embedded BASIC with $Ba^{2+}$ were collected using a Titan Krios (300 keV) microscope (Thermo Fisher) with a Falcon4i direct electron detector at HHMI Janelia Research Campus. In EER format, data acquisition was acquired automatically using serialEM across a defocus range of −0.8 to −2.2 μm, a total dose of 50 e⁻/Å$^2$, and a physical pixel size of 0.743 Å/pix.

## Image processing

The movie frames were imported into CryoSPARC v4 and underwent motion correction and contrast transfer function (CTF) refinement using Patch Motion Correction and Patch Multi CTF estimation, respectively[60]. Corrected movies with CTF values exceeding 6 Å, as well as outliers in ice thickness, average defocus, and motion were manually excluded. Particle picking commenced with Blob picker, employing size cutoffs ranging from 100 Å to 120 Å. A subset of extracted particles underwent initial cleaning via 2D classification before proceeding to ab initio reconstruction with four classes. The classes resembling hBASIC were selected to generate templates for template picking. Particles selected from both blob and template picking were extracted with a box size of 400 pixels, binned to 100 pixels, and combined, with duplicate particles removed using the 'Remove Duplicate Particles' function. Utilizing the four ab initio reconstructions as input volumes, particles underwent several rounds of heterogeneous refinement (C3 symmetry). Selected particles were re-extracted at a box size of 200 pixels and after extraction the particles were subjected to ab initio reconstruction of 4 classes, followed by heterogeneous refinement (C3 symmetry). Final particle stack was re-extracted to a box size of 400 pixels before undergoing further 3D classification.

Unbinned particles of hBASIC prepared in the presence of calcium or barium were initially subjected to ab initio classification with four classes using default settings. Selected particles from one class underwent further refinement through two subsequent ab initio jobs. The first job, employing two classes with a class similarity of 0.1, aimed to eliminate remaining preferred orientation, while the second job, utilizing two classes with a class similarity of 0.9, targeted the removal of any undesirable particles. The selected particles then underwent 2D classification to manually discard those with apparent low signal-to-noise ratio. The final particles, along with the map generated from the previous ab initio steps, were refined using non-uniform refinement[61] (C3 symmetry). This refined map underwent additional refinement steps including local refinement, local and global CTF refinement, and a final non-uniform refinement (C3 symmetry).

Similarly, unbinned particles of hBASIC prepared in the presence of EGTA were subjected to ab initio classification with four classes and default settings. The selected class particles and map were refined using non-uniform refinement. To enhance the features of the transmembrane domain, conformations were sorted using 3D variability analysis with a focus on the transmembrane domain. This analysis yielded two distinct classes, which were subsequently refined separately via non-uniform refinement (C3 symmetry). The resulting maps underwent multiple rounds of local refinement, followed by local and global CTF refinement, and a final non-uniform refinement (C3 symmetry).

## Model building, refinement, and pore analysis

The initial hBASIC model was constructed utilizing the predicted model from AlphaFold2[62], which was then fitted into the unsharpened map of hBASIC under 2 mM calcium conditions using UCSF ChimeraX (v1.8)[63]. Manual adjustments and refinement of the model were carried out in Coot (v0.9.8.6)[64]. Subsequent refinements were performed iteratively employing Coot, Isolde (v1.6.0)[65], and real-space refinement in Phenix (v1.18)[66]. Final models stereochemistry and geometry were evaluated using MolProbity (v4.5.2)[67]. N-glycosylation was modelled using the carbohydrate module in Coot. The model under 2 mM calcium conditions served as the starting model for structures in the presence of EGTA. Pore calculations were conducted using HOLE[68] within Coot. Visualization and preparation of figures was done in UCSF ChimeraX and Pymol (v2.5.5) (https://pymol.org).

## Two-electrode voltage clamp (TEVC) electrophysiology

**Oocytes.** *Xenopus laevis* oocytes were purchased from Ecocyte Biosciences, and arrived defolliculated and in Barth's solution (88 mM NaCL, 1 mM KCl, 0.82 mM MgSO4, 0.33 mM Ca(NO$_3$)$_2$, 0.41 mM CaCl$_2$, 2.4 mM NaHCO$_3$, 5 mM Tris-HCl) supplemented with penicillin (100 U/mL) and Streptomycin.

**RNA preparation and injection.** The hBASIC gene was subcloned into the pcDNA3.1 vector with no GFP, protease sites, or affinity tags present. Mutagenesis was performed to the initial WT construct with primers listed in Supplementary Table 1 using CloneAmp HiFi PCR Premix (Takara). The plasmid was digested using KasI and was used for RNA preparation using mMessage mMachine T7 Ultra transcription kit (Invitrogen). RNA was analyzed using agarose gel electrophoresis prior to being aliquoted into 2 µL portions and flash frozen. RNA was stored at −80 °C for no longer than 3 months. *Xenopus* oocytes were purchased from Ecocyte Biosciences and were injected with 8 ng of RNA immediately upon arrival. During expression, oocytes were incubated at 19 °C in a ND96 solution (in mM: 96 NaCl, 2 KCl, 1.8 CaCl$_2$, 1 MgCl$_2$, 5 HEPES pH 7.4).

**Proteoliposome preparation and injection**. Brain total lipids (Avanti Polar Lipids) dissolved in chloroform were subjected to a vacuum desiccator overnight. The lipids were resuspended into TBS buffer (20 mM Tris-HCl pH 8.0, 150 mM NaCl) by iterative freezing with liquid nitrogen and thawing via sonication. To achieve uniformly sized liposomes, the resuspended material was extruded back and forth through a double layer of 100 nm filters positioned between two syringes.

The extruded liposomes were then isolated by ultracentrifugation, and the resulting pellet was subsequently resuspended in 150 µL of protein solution. For each reaction, 250 µg of freshly purified hBASIC protein was directly added to 12 mg of liposome pellet. To disperse the liposomes, 5 µL of 50 mM DDM was added to the mixture, followed by an incubation on ice for 30 minutes.

To remove detergent and facilitate spontaneous reconstitution of hBASIC into liposomes, three rounds of 500 mg of washed Biobeads were added prior to 3-h incubations at 4 °C with gentle agitation. Subsequently, the resulting proteoliposomes were combined and then isolated by ultracentrifugation. This pellet was then resuspended in 200 µL TBS buffer to achieve a concentration of 1.5 mg/mL BASIC. For proteoliposome injections, 1 µL of solution containing 300 µg of hBASIC was diluted with 100 µL of TBS buffer to create a final concentration of 3 µg. A volume of 48.6 nL of the diluted stock was injected into oocytes and the oocytes were then incubated at 4 °C in ND96 solution overnight. Following this incubation period, the oocytes were used for recording.

**TEVC recordings and analysis.** Data was acquired from oocytes at room temperature using Axon CNS Molecular Devices Axoclamp 900 A amplifier and the Clampex 10.3 software. Superfusion of oocytes was conducted utilizing a Rapid Solution Changer 160 (BioLogic) multi-channel perfusion system. Unless specified otherwise, the current was measured at a holding potential of −70 mV. Data were filtered at 20 Hz and acquired at a rate of 1 kHz. Current amplitudes were measured using Clampfit 10.7.1. The standard bath solution for experiments consisted of 140 mM NaCl, 1 mM CaCl$_2$, 10 mM HEPES pH 7.4, and was supplemented with 100 µM Flufenamic acid (FFA) (Sigma) to inhibit conductance of oocytes in divalent free media[8]. For Ca$^{2+}$ concentrations lower than 1 mM, the appropriate amount of CaCl$_2$ and EGTA was added using Ca-EGTA Calculator v1.3[69]. For maximum current acquisition, Ca$^{2+}$ was substituted with EGTA.

**Inhibition dose response (IC$_{50}$) experiments.** To determine the half-maximal inhibitory concentration, an episodic protocol was designed in Clampex to apply discrete concentrations of the antagonist, Ca$^{2+}$, for 20-second durations. For IC$_{50}$ measurements involving Mg$^{2+}$, Ba$^{2+}$, or Sr$^{2+}$, that respective divalent ion replaced Ca$^{2+}$ in the protocol. The protocol began by establishing the baseline and maximum current

using a "standard bath" and a "Ca²⁺-free bath," respectively. The standard bath contained 1 mM Ca²⁺ for WT experiments or 10 mM Ca²⁺ for mutant and divalent ion experiments, while the Ca²⁺-free bath contained no added Ca²⁺ and was supplemented with 200 μM EGTA to chelate ambient Ca²⁺. The dilution series started with the application of the standard bath, followed by test concentrations of Ca²⁺ in descending order, ending with the Ca²⁺-free bath. The protocol concluded with a final application of the standard bath to re-establish baseline current. Unless otherwise indicated in the main text, the protocol was performed on at least six distinct oocytes (N = 6) per condition (WT, mutants, and pH). For each Ca²⁺ concentration, the sustained current averaged over the 20-second recordings was baseline-subtracted. Within each group, currents were normalized to the maximum current determined for the individual oocyte. The data were fit to a four-parameter sigmoidal curve using the "[inhibitor] vs. response–variable slope" function in GraphPad Prism 10 to calculate $IC_{50}$ values. Mean $IC_{50}$ values with standard error of the mean (SEM) are presented in the main text.

**Current-voltage experiments.** Using an episodic stimulation protocol that was designed in Clampex, the current-voltage relationship of BASIC in the presence of 10 mM, 1 mM, and 10 μM Ca²⁺, and the absence of Ca²⁺ was measured. The protocol consisted of 13 sweeps, with each sweep beginning with the baseline holding potential of −70 mV for 300 ms durations prior to application of a voltage-test application for 600 ms. The voltage-test application started at −40 mV and increased each sweep by 10 mV, with the final sweep being +80 mV. The sustained current was averaged over the 600 ms recording. For each different [Ca²⁺], each voltage-test application was replicated 6 times, amongst 6 distinct oocytes. The resulting data was subtracted by control data from uninjected oocytes (Supplementary Fig. 15a) before being graphed in Prism 10.

To verify that endogenous Ca²⁺-activated chloride channels in oocytes were not impacting the interpretations of the current-voltage experiments, we repeated these experiments with Ba²⁺, an effective blocker of these endogenous channels, in place of Ca²⁺[70]. We did not find a difference in IV curves of hBASIC in the presence of Ba²⁺, in comparison to Ca²⁺ (Supplementary Fig. 15b).

### Reporting summary
Further information on research design is available in the Nature Portfolio Reporting Summary linked to this article.

## Data availability
The cryo-EM maps and coordinates for the hBASIC structures in order of Ca²⁺-closed, EGTA-open, and EGTA-closed, have been deposited in the Electron Microscopy Data Bank (EMDB) under accession numbers EMD-48342, EMD-48343, EMD-48380, and in the Protein Data Bank (PDB) under accession codes 9MKY, 9MKV, 9MLV, respectively. Ba²⁺-closed map has been deposited in EMDB under accession number EMD-48570. The maps within these depositions include both half maps, sharpened, and the unsharpened maps used for model building and refinement. Source data are provided with this paper.

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

## Acknowledgements

We thank L. Anson and M.L. Mayer for their feedback and edits to the manuscript, R. Courtney for help with manuscript preparation, and the Gouaux and Baconguis lab members for their helpful feedback and discussion, especially K. Hartfield. Grid screening for electron microscopy was carried out at the Multiscale Microscopy Core (MMC) at OHSU, with special thanks to C. López for providing microscope access and assistance. A portion of this research was supported by NIH grant R24GM154185 and performed at the Pacific Northwest Center for Cryo-EM (PNCC) with assistance from J. Myers. We acknowledge use of the cryo-EM facility at the HHMI Janelia research campus. The research was supported by the National Institute of Neurological Disorders and Stroke (F31NS120713 to M.M.F. and 5R01NS038631 to E.G.) and National Science Foundation (NSFGRFP-1000271223 to M.M.F.). Additional support was provided by ARCS Foundation to M.M.F. E.G. is an investigator of the Howard Hughes Medical Institute and thanks Bernard and Jennifer LaCroute for generous support.

## Author contributions

M.M.F. and E.G. designed the project. M.M.F. performed the research and data analysis. M.M.F. wrote the manuscript, and all authors edited the manuscript.

## Competing interests

The authors declare no competing interests.
