## [Transparent Peer Review file · Nature Communications]

The bile acid-sensitive ion channel is gated by Ca^{2+} -dependent conformational changes in the transmembrane domain

Corresponding Author: Dr Eric Gouaux

Version 0:

Reviewer comments:

Reviewer #1

(Remarks to the Author)

The manuscript by M.M. Freitas and E. Gouaux presents the high-resolution cryo-EM structure of the human bile acid-sensitive ion channel (BASIC) in Ca^{2+} -free and Ca^{2+} -bound states. The study reveals a previously unknown mechanism where the presence or absence of Ca^{2+} in the pore affects the conformation of the channel rather than only blocking the pore, thus unravelling a putative new regulatory mechanism of BASIC. Electrophysiological recordings performed in *Xenopus laevis* oocytes suggest an effect of the membrane potential on the inhibition of BASIC by Ca^{2+} . The work employs state-of-the-art methodologies and represents a significant advancement in understanding the regulation of BASIC and possibly other ASICs.

There are a few concerns that should be addressed before publication.

Major

1. BASIC is sensitive to its membrane environment and for mBASIC different electrophysiological properties were described depending on the cell system used for expression (Lenzig et al. 2019). In the study presented here, BASIC integrity after isolation from Hek cells was tested by introducing purified BASIC into proteoliposomes followed by injection into *Xenopus* oocytes and TEVC recordings. It is possible, if not likely, that proper BASIC integrity is lost during the purification process and only restored upon insertion into the membrane of *Xenopus* oocytes. This would question the relevance of the structure presented here. While it is impossible to test the electrophysiological properties of BASIC in nanodisks, we think that it would be more convincing if a second control experiment would be performed where BASIC function (in proteoliposomes) is tested in another cell system with different membrane composition e.g. Hek or cos7 cells.
2. In *Xenopus* oocytes strong depolarization above +40 to +50 mV often induces Na^{+} currents that could contaminate the currents observed in the study presented here (e.g. Vasilyev and Rakowski, 2001 and others). Thus, the current-voltage relationships observed here (fig. 5c) are likely contaminated with BASIC-independent current(s) and the voltage-dependence of the Ca^{2+} block may be overestimated. This can easily be tested and estimated by measuring uninjected or water-injected oocytes.
3. The authors compare the newly obtained BASIC structure with resting and desensitized structures of cASIC1a (lines 141-161, Supp. Fig. 5) and correctly pinpoint the structural difference in the finger domain. However, the differences in the thumb domain are due to ASIC1a's ability to undergo desensitization, which does not occur in BASIC. In Supp. Fig. 5a the authors compare the rmsd of hBASIC and the desensitized cASIC1a. Here a comparison with the resting state of ASIC1a is more suitable. Please add or replace Supp. Fig. 5a with this analysis. This should also be reflected in the text.
4. In line 216-217 the authors describe "a tyrosine residue (Y100) that inserts itself into the central vestibule, adjacent to L425". This is an important finding, but the related illustration (Fig. 2b) does not show this properly. Please optimize the figure to highlight this finding.
5. Fig. 3a-b display the pore of hBASIC. However, only the pore profile (Fig. 3a) and the architecture of the pore (Fig. 3b) for

the closed state of the channel are shown. Please also add the pore profile of the open state (add to Fig. 3a) and show a superimposed figure of the closed and open pore (Fig. 3b) to underline the differences of both conformations.

Minor

1. The cryo-EM experiment without Ca²⁺ revealed two structures of BASIC termed EGTA-closed and EGTA-opened. Lefèvre et al. (2014) also showed that hBASIC exists in an open and closed state in the presence of the activator deoxycholic acid. This should be mentioned earlier and discussed in more detail.
2. rBASIC contains an amphiphilic helix in the N-terminal domain, possibly interfering with the conformation of the channel. Even though the intracellular domains are not resolved in this study because a possible relation to the results presented in this study should be discussed.
3. Page 3, line 64 “high millimolar quantities”, should read “high millimolar concentrations”.
4. On page 3, line 68, the authors state that even a small reduction in Ca²⁺o reduction leads to an increase in BASIC activity. According to the literature and the data presented here the IC₅₀ for Ca²⁺o is 7 μM. The term “small reduction” does not seem to be adequate in this context as reduction of Ca²⁺o by more than a factor of 10 is required to induce a “small” (10%) increases of BASIC activity.
5. Please add n-numbers to supplementary figures where appropriate.
6. Fig. 1b: The GAS-belt in this figure is barely visible and hard to identify. Please enhance its visibility.
7. Please revise the sentence in lines 316–317 to eliminate redundant word choices and improve clarity.

Reviewer #2

(Remarks to the Author)

BASIC, the least studied member of the ENaC superfamily, plays crucial roles in numerous physiological processes. In this study, the author conducts a comprehensive structural and functional analysis of BASIC. They elucidate the overall structure of human BASIC and provide insights into the mechanism of calcium inhibition. Notably, calcium binds at the top of the pore, and structural comparisons uncover novel calcium-dependent conformational changes within the ENaC family. Overall, the conclusions are well-supported, and the structural and functional studies are solid and rigorous.

I have several minor issues.

Calcium removal results in significant conformational changes in BASIC, but does not necessitate a similar conformational alteration in ASIC3. What factors could be responsible for this difference?

How is the signal of calcium binding or unbinding relayed to the β1-β2 and β11-β12 linkers? Do the β strands connecting the transmembrane helices to the β1-β2 and β11-β12 linkers undergo any structural changes?

As mentioned in the introduction, a missense mutation in BASIC is associated with pregnancy loss. It would be interesting to study and discuss how this mutation affects the protein's function and contributes to the development of the disease.

Lines 205-206 and 216 mention the interactions between β1-β2 and β11-β12. It would be helpful to include a figure that visually represents this interaction.

Alongside Fig2d, including an additional figure that shows a bottom or top view of the trimer may better illustrate the movements of the helices resulting in pore expansion.

Reviewer #3

(Remarks to the Author)

Deg/ENaC superfamily encompasses ion channels activated by very diverse stimuli (protons, neuropeptides, mechanical force, bile acids, etc.). To date, the structural studies of this family have mostly focused on the acid sensing ion channels (ASICs). The manuscript by Freitas and Gouaux reveals the first structures of the human bile acid-sensitive ion channel (BASIC) in resting and putatively conductive conformations, with and without the inhibiting ion Ca²⁺. The Ca²⁺-binding site observed in the structures is located at the extracellular entrance to the pore. The structures are complemented by functional characterization of the wild-type channel and various mutants, probing the proposed activation mechanism. The authors have also solved the structures of BASIC in the presence of Ba²⁺ to verify the location of the Ca²⁺-binding site.

The study proposes a novel activation mechanism, where BASIC is inhibited by the divalent ion (Ca²⁺) binding at the extracellular entrance to the pore, while removing Ca²⁺ triggers the pore expansion, with other conformational changes propagating to the extracellular domain. This is in contrast to other Deg/ENaCs characterized to date, where ligand binding triggers the changes in the extracellular domain, which are further propagated to the channel pore.

The work will be of great interest to the Deg/ENaC field, and more broadly to those interested in the mechanisms channel gating and ion conduction, the methodology used is appropriate and the data support the conclusions. The manuscript is well written, and the figures are very clear. I would also like to commend the authors on their efforts to show that the

detergent-solubilized protein retains its function. Below are my comments.

Essential:

1. Please include the processing workflow for the Ba²⁺ dataset and the map validation, it is important to demonstrate how it compares to the Ca²⁺ dataset. In addition to Fig. 5b, the authors should also include a figure where the surrounding density is shown at the same contour as Ca²⁺/Ba²⁺, and it would also be informative to include the EGTA maps/models in this comparison to clearly demonstrate the presence/absence of densities, and the quality of the maps in that region in general. This is particularly important as Deg/ENaC maps often suffer from worse resolution in the pore region.

Minor:

1. Not very clear from the figures whether the conformations of E440 sidechain are ambiguous or not (given the lower resolution in the pore region, and commonly occurring radiation damage to the acidic sidechains), could the authors comment and illustrate this better? If these sidechains are not visible, perhaps it is better to interpret the conformational transitions as backbone movements only (concerns figures 3c, 4c).
2. Perhaps injecting oocytes with empty liposomes is a better control for suppl. fig. 2d
3. Angular distribution plots in suppl. fig. 3c, 6c look like the maps were not symmetrized, while according to the processing workflows final maps are C3-symmetric. In suppl. fig. 6c the plots appear to be duplicated for the two final maps.
4. Suppl. fig. 6d, please label the final maps which is which

Typos and other corrections:

1. Lines 396 – 397, some repetition with final concentrations
2. Line 450, “collected on a Krios?”

Version 1:

Reviewer comments:

Reviewer #1

(Remarks to the Author)

The author's efforts are appreciated. All questions raised were addressed adequately.

Thank you for taking the time to review our manuscript. Below, in blue text, we respond to the reviewers' comments:

Reviewer #1 (Remarks to the Author):

The work employs state-of-the-art methodologies and represents a significant advancement in understanding the regulation of BASIC and possibly other ASICs.

There are a few concerns that should be addressed before publication.

We appreciate the time and expertise you dedicated to reviewing our manuscript and for your helpful suggestions to enhance its scientific merit.

Major

1. BASIC is sensitive to its membrane environment and for mBASIC different electrophysiological properties were described depending on the cell system used for expression (Lenzig et al. 2019). In the study presented here, BASIC integrity after isolation from Hek cells was tested by introducing purified BASIC into proteoliposomes followed by injection into *Xenopus* oocytes and TEVC recordings. It is possible, if not likely, that proper BASIC integrity is lost during the purification process and only restored upon insertion into the membrane of *Xenopus* oocytes. This would question the relevance of the structure presented here. While it is impossible to test the electrophysiological properties of BASIC in nanodiscs, we think that it would be more convincing if a second control experiment would be performed where BASIC function (in proteoliposomes) is tested in another cell system with different membrane composition e.g. Hek or cos7 cells.

We appreciate these comments and are particularly aware of the sensitivity of ASIC family members to detergent and membrane environment. While no experiment is perfect, we suggest that the oocyte experiments we have carried out demonstrate that structural integrity of the TM region of hBASIC was not lost irreversibly during purification. We also suggest that the reconstitution of hBASIC in nanodiscs is a reasonable approximation to a *bona fide* membrane environment. Lastly, while ideally we would carry out the suggested experiments of introducing hBASIC in proteoliposomes into cells with membranes different from oocytes, such as those of HEK or Cos7 cells, there are not robustly established procedures for doing such experiments and carrying them out on hBASIC is an independent research endeavor in-and-of itself and, as such, is well beyond the scope of the current work.

2. In *Xenopus* oocytes strong depolarization above +40 to +50 mV often induces Na⁺ currents that could contaminate the currents observed in the study presented here (e.g. Vasilyev and Rakowski, 2001 and others). Thus, the current-voltage relationships observed here (fig. 5c) are likely contaminated with BASIC-independent current(s) and the voltage-dependence of the Ca²⁺ block may be overestimated. This can easily be tested and estimated by measuring uninjected or water-injected oocytes.

Thank you for bringing this to our attention. Based on your suggestions, we have subtracted contaminating currents observed in uninjected oocytes from the recordings of hBASIC injected

oocytes. Subtracted currents showed no changes in outward rectification of hBASIC in the presence of calcium. While there is a small Na⁺ current in uninjected oocytes observed above +40mV, the max current is far lower than the hBASIC injected oocytes. For example, the largest current observed in an average control oocyte in the presence of 10mM calcium is +527 nA at +80mV. Upon subtraction of this current from averaged hBASIC injected oocytes, the average current is +6,073.8 nA at +80mV. With the elimination of contaminated currents, hBASIC currents in the presence of calcium exhibit an outward rectification. In the revised Supplementary Fig. 15, we have added a figure (Supplementary Fig. 5a) showing current-voltage relationship observed in uninjected oocytes.

3. The authors compare the newly obtained BASIC structure with resting and desensitized structures of cASIC1a (lines 141-161, Supp. Fig. 5) and correctly pinpoint the structural difference in the finger domain. However, the differences in the thumb domain are due to ASIC1a's ability to undergo desensitization, which does not occur in BASIC. In Supp. Fig. 5a the authors compare the rmsd of hBASIC and the desensitized cASIC1a. Here a comparison with the resting state of ASIC1a is more suitable. Please add or replace Supp. Fig. 5a with this analysis. This should also be reflected in the text.

We appreciate the reviewer's insightful feedback and have added an analysis comparing the RMSD of hBASIC to the resting state of cASIC1a in Supplementary Fig. 5a, as suggested. We believe that displaying both strengthens the interpretation of hBASIC's structural differences. Our initial expectation was that the Ca²⁺-closed conformation of hBASIC would resemble the resting state of cASIC1a, since as you had mentioned, hBASIC does not desensitize. However, we were surprised to find that hBASIC exhibited a greater structural overlap overall with the cASIC1a desensitized state. In lines 129-132, we discuss the RMSD comparison between BASIC and both the resting and desensitized states of cASIC1a. Because the RMSD is lower for the desensitized state, indicating higher structural conservation overall, we initially selected the global alignment to this conformation. As the exception to the greater conservation is at the thumb domain, as correctly pointed out, we compared the secondary structure of this region to both cASIC1a desensitized and resting in Supplementary Fig. 5b.

4. In line 216-217 the authors describe "a tyrosine residue (Y100) that inserts itself into the central vestibule, adjacent to L425". This is an important finding, but the related illustration (Fig. 2b) does not show this properly. Please optimize the figure to highlight this finding.

We thank the reviewer for this suggestion and have provided additional figures in the supplement that show the interactions between the β -linkers. Supplementary Fig. 10a-b highlights the placement of Y100 in proximity to L425 in the Ca²⁺-closed state and EGTA-open state.

5. Fig. 3a-b display the pore of hBASIC. However, only the pore profile (Fig. 3a) and the architecture of the pore (Fig.3b) for the closed state of the channel are shown. Please also add the pore profile of the open state (add to Fig. 3a) and show a superimposed figure of the closed and open pore (Fig. 3b) to underline the differences of both conformations.

To complement the HOLE profile analysis for Ca²⁺-closed, EGTA-open, and EGTA-closed in Fig

3b, we have included the corresponding cartoon representation of the pore profile in all three states in the supplement. Supplementary Fig. 11 compares the hole profile analysis across these conformations, providing a clearer visualization of the structural differences.

Minor

1. The cryo-EM experiment without Ca²⁺ revealed two structures of BASIC termed EGTA-closed and EGTA-opened. Lefèvre et al. (2014) also showed that hBASIC exists in an open and closed state in the presence of the activator deoxycholic acid. This should be mentioned earlier and discussed in more detail.

The single channel data from Lefèvre et al. (2014) in the presence of high and low calcium guided the interpretation of our structural states, as discussed in lines 124-127 and 169-173. While the single channel analysis of BASIC in the presence of the activator deoxycholic acid is an interesting and important finding, discussion of single channel data of BASIC in the presence of deoxycholic acid may cause confusion to the reader as we do not use this activator in this study. Future structural studies of BASIC in complex with deoxycholic acid may be the best place to discuss the single channel data from Lefèvre et al. (2014).

2. rBASIC contains an amphiphilic helix in the N-terminal domain, possibly interfering with the conformation of the channel. Even though the intracellular domains are not resolved in this study because a possible relation to the results presented in this study should be discussed.

We thank the reviewer for this suggestion. We have incorporated this exciting finding into the discussion (lines 374-376), providing additional context and strengthening the overall interpretation of our results.

Like Ca²⁺, these membrane-active substances may influence channel gating by perturbing the conformation of the TMD, thereby indirectly modifying the β -linkers. Supporting this model, prior studies on rat BASIC demonstrates that a cytosolic α -helix regulates channel activity and mediates its sensitivity to cholesterol, with removal of the α -helix or depletion of cholesterol resulting in channel hyperactivity [8]. It is likely that the combined interplay between membrane-based substances and Ca²⁺ serves to fine-tune channel activity in vivo. However, further research into interactions of membrane substances with BASIC and ASIC will be needed to uncover the mechanism of their membrane sensitivity.

3. Page 3, line 64 “high millimolar quantities”, should read “high millimolar concentrations”.

We have amended line 64.

4. On page 3, line 68, the authors state that even a small reduction in Ca²⁺ leads to an increase in BASIC activity. According to the literature and the data presented here the IC₅₀ for Ca²⁺ is 7 μ M. The term “small reduction” does not seem to be adequate in this context as reduction of Ca²⁺ by more than a factor of 10 is required to induce a “small” (10%) increases of BASIC activity.

We have revised line 68.

5. Please add n-numbers to supplementary figures where appropriate.

Thank you for bringing this to our attention. n-numbers have been added to the figure legends of supplementary figures, where appropriate.

6. Fig. 1b: The GAS-belt in this figure is barely visible and hard to identify. Please enhance its visibility.

We have adjusted the labeling to make it easier to identify.

7. Please revise the sentence in lines 316–317 to eliminate redundant word choices and improve clarity.

Lines 316-317 have been revised.

Reviewer #2 (Remarks to the Author):

I have several minor issues.

Calcium removal results in significant conformational changes in BASIC, but does not necessitate a similar conformational alteration in ASIC3. What factors could be responsible for this difference?

To our knowledge, the structure of ASIC3 has not been resolved. Based on interpretation of functional data of ASIC3, we speculate the differences between BASIC and ASIC3 are due to the presence of an additional calcium binding site in the acidic pocket of ASIC3. We speculate about the factors responsible for the differences between ASIC3 and BASIC in lines 333-347 of the discussion.

How is the signal of calcium binding or unbinding relayed to the β 1- β 2 and β 11- β 12 linkers? Do the β strands connecting the transmembrane helices to the β 1- β 2 and β 11- β 12 linkers undergo any structural changes?

We have included Supplementary Fig. 9a-b, which illustrates how the "molecular clutch", the β -linkers, are structurally connected to the TMD via the β -strands (β 1 and β 12) and the wrist regions. This figure highlights the structural relationship between these elements and how they may facilitate the transmission of conformational changes upon calcium binding or unbinding.

As mentioned in the introduction, a missense mutation in BASIC is associated with pregnancy loss. It would be interesting to study and discuss how this mutation affects the protein's function and contributes to the development of the disease.

We agree with you that the missense mutation in BASIC that is associated with pregnancy loss is indeed very interesting. However, to address how this mutation affects the protein's function would require thorough experiments that we believe are beyond the scope of this current study.

Lines 205-206 and 216 mention the interactions between β 1- β 2 and β 11- β 12. It would be helpful to include a figure that visually represents this interaction.

To provide further representation of this interaction, we have added Supplementary Fig. 9 that shows further interaction between β 1- β 2 and β 11- β 12.

Alongside Fig2d, including an additional figure that shows a bottom or top view of the trimer may better illustrate the movements of the helices resulting in pore expansion.

We thank the reviewer for this thoughtful suggestion. To better illustrate the movements of the helices leading to pore expansion, we have included Supplementary Fig. 9c-d, which provides both bottom and top views of the trimer. These views help to more clearly visualize the conformational changes associated with channel gating. We agree that this addition strengthens the presentation of our findings.

Reviewer #3 (Remarks to the Author):

Essential:

1. Please include the processing workflow for the Ba²⁺ dataset and the map validation, it is important to demonstrate how it compares to the Ca²⁺ dataset. In addition to Fig. 5b, the authors should also include a figure where the surrounding density is shown at the same contour as Ca²⁺/Ba²⁺, and it would also be informative to include the EGTA maps/models in this comparison to clearly demonstrate the presence/absence of densities, and the quality of the maps in that region in general. This is particularly important as Deg/ENaC maps often suffer from worse resolution in the pore region.

We appreciate the reviewer's suggestion and have addressed this by adding Supplementary Fig. 12, which details the processing workflow for the Ba²⁺ dataset along with map validation, facilitating a direct comparison with the Ca²⁺ dataset.

To further illustrate the presence or absence of ion density, we have also included Supplementary Fig. 13, which presents the ion densities and surrounding density for the Ca²⁺-closed, Ba²⁺-closed, EGTA-open, and EGTA-closed maps at the same contour level. This comparison highlights both map quality in the pore region and differences in ion occupancy across these conditions.

We agree with you that Deg/ENaC channel maps often exhibit lower resolution in the pore region, a challenge also observed in hBASIC. In particular, the lower half of the pore frequently requires 3D classification and local refinement to enhance resolution. As shown in Supplementary Fig. 12, the Ba²⁺-closed map is anisotropic. However, as our primary objective was to compare ion occupancy at the top of the pore, where resolution was already sufficient for this analysis, we prioritized optimizing the ion-binding region rather than further refining resolution in the lower half of the pore.

Minor:

1. Not very clear from the figures whether the conformations of E440 sidechain are ambiguous or not (given the lower resolution in the pore region, and commonly occurring radiation damage to the acidic sidechains), could the authors comment and illustrate this better? If these sidechains are not visible, perhaps it is better to interpret the conformational transitions as backbone movements only (concerns figures 3c, 4c).

We appreciate the reviewer's concern regarding the resolution and potential radiation damage affecting the visibility of E440 sidechain conformations. In the Ca²⁺-closed map, the acidic residues are well-resolved; however, at equivalent contour levels, the conformation of these residues are ambiguous in the EGTA-closed and EGTA-open maps. As a result, we interpreted the conformational transitions as backbone movements.

To clarify this, we have updated the text and Fig. 4c. Regarding Fig. 3c, the calculation of pore expansion is based solely on the backbone conformation. The residues were colored in the

figure only to visually illustrate pore expansion, rather than to suggest that the sidechains themselves were explicitly resolved.

2. Perhaps injecting oocytes with empty liposomes is a better control for suppl. fig. 2d.

We appreciate the reviewer's suggestion to use empty liposome-injected oocytes as a control in Supplementary Fig. 2d. While this would be a useful additional control, we believe that our current approach – using uninjected oocytes, which show no current, along with inhibition by the hBASIC specific blocker, diminazene – effectively demonstrates that the observed currents are specific to the expressed channel. Given that uninjected oocytes exhibit no detectable current, we expect that empty liposome injection would yield similar results. However, we acknowledge the reviewer's point and will consider this control in future studies.

3. Angular distribution plots in suppl. fig. 3c, 6c look like the maps were not symmetrized, while according to the processing workflows final maps are C3-symmetric. In suppl. fig. 6c the plots appear to be duplicated for the two final maps.

The datasets were indeed symmetrized. To ensure accuracy, we re-downloaded the angular distribution plots directly from the original CryoSPARC jobs and replaced the images in Supplementary Fig. 3c and 6c. Additionally, we updated the plots in Fig. 6c to eliminate any potential duplication. It does appear that the FSC plot for EGTA-closed in Supplementary Fig. 7 was mistakenly duplicated, and we have now replaced it with the correct version. We appreciate the reviewer for bringing this to our attention.

4. Suppl. fig. 6d, please label the final maps which is which

We have now labeled the final maps in Supplementary Fig. 6d to clearly indicate which is which.

Typos and other corrections:

1. Lines 396 – 397, some repetition with final concentrations

Thank you for pointing out the repetition. We have revised that paragraph to remove redundant mentions of concentrations.

2. Line 450, "collected on a Krios?"

Thank you for catching this extra sentence, it has been removed from the text.